# Subcellular Effectors of Cocaine Cardiotoxicity: All Roads Lead to Mitochondria—A Systematic Review of the Literature

**DOI:** 10.3390/ijms241914517

**Published:** 2023-09-25

**Authors:** Michela Peruch, Emiliana Giacomello, Davide Radaelli, Monica Concato, Riccardo Addobbati, Alessandra Lucia Fluca, Aneta Aleksova, Stefano D’Errico

**Affiliations:** 1Department of Medical, Surgical and Health Sciences, University of Trieste, 34149 Trieste, Italy; michela.peruch@studenti.units.it (M.P.); egiacomello@units.it (E.G.); davide_radaelli@hotmail.it (D.R.); monica.concato@studenti.units.it (M.C.); alessandralucia.fluca@units.it (A.L.F.); aaleksova@units.it (A.A.); 2Institute for Maternal and Child Health, IRCCS “Burlo Garofolo”, 34137 Trieste, Italy; riccardo.addobbati@burlo.trieste.it; 3Cardiothoracovascular Department, Azienda Sanitaria Universitaria Giuliano Isontina, 34149 Trieste, Italy

**Keywords:** cocaine, cardiotoxicity, apoptosis, mitochondria impairment, oxidative stress

## Abstract

Cocaine abuse is a serious public health problem as this drug exerts a plethora of functional and histopathological changes that potentially lead to death. Cocaine causes complex multiorgan toxicity, including in the heart where the blockade of the sodium channels causes increased catecholamine levels and alteration in calcium homeostasis, thus inducing an increased oxygen demand. Moreover, there is evidence to suggest that mitochondria alterations play a crucial role in the development of cocaine cardiotoxicity. We performed a systematic review according to the Preferred Reporting Items for Systemic Reviews and Meta-Analysis (PRISMA) scheme to evaluate the mitochondrial mechanisms determining cocaine cardiotoxicity. Among the initial 106 articles from the Pubmed database and the 17 articles identified through citation searching, 14 final relevant studies were extensively reviewed. Thirteen articles included animal models and reported the alteration of specific mitochondria-dependent mechanisms such as reduced energy production, imbalance of membrane potential, increased oxidative stress, and promotion of apoptosis. However, only one study evaluated human cocaine overdose samples and observed the role of cocaine in oxidative stress and the induction of apoptosis though mitochondria. Understanding the complex processes mediated by mitochondria through forensic analysis and experimental models is crucial for identifying potential therapeutic targets to mitigate or reverse cocaine cardiotoxicity in humans.

## 1. Introduction

Drug abuse represents a major public health problem worldwide [1]. In 2020, the European Monitoring Center for Drugs and Drug Addiction (EMCDDA) highlighted that 13.4% of overdose deaths involved cocaine [2]. In Europe, after a short phase of decreased use of cocaine in 2020, its consumption dramatically returned to pre-pandemic levels already in 2021 [2].

Cocaine is produced from the leaves of the Erythroxylon coca plant in different formulations, which translate in various ways of administration [3]. Typically, powder cocaine is snorted, but it can also be taken orally or injected. Crack, which is produced as rocks from cocaine hydrochloride with baking soda or ammonium, is smoked [3]. Cocaine is well absorbed by all routes of administration, differing in time to peak in cerebral circulation and duration of effects [3]. In particular, intravenous administration and inhalation produce a bioavailability of the drug greater than 90% [3]. However, while intravenous administration is associated with an intense onset of effects in about 5 min, insufflation and smoking, which are the most commonly used, allow for a response after one to three minutes, on average [3]. On the contrary, snorting is characterized by a bioavailability of less than 80%, a delayed plasma peak, due to its vasoconstrictive properties, and a long-lasting effect in the range of 15–30 min [3]. 

The metabolism of cocaine implies the hydrolysis of the two ester moieties: the alkyl ester, which is hydrolyzed to benzoylecgonine by liver methylesterases and spontaneous hydrolysis, and the phentyl ester, which is hydrolyzed to ecgonine methyl ester by cholinesterase and liver benzoylesterases [3,4]. The metabolites of cocaine are excreted in the urine [3,4].

Clinically, the mechanism of action of cocaine is characterized by a decrease in the permeability of axonal membrane to sodium, which results in an inhibition of initiation and conduction of nerve impulses. Cocaine blocks the presynaptic reuptake of norepinephrine, dopamine (DA), and serotonin (5-HT), thus causing a potent sympathomimetic effect [4]. Above all, cocaine binds to sodium-binding site and alters chloride-binding sites on the DA transporter, inhibiting the translocation of DA across the presynaptic neuronal membrane. This causes an increased extracellular DA concentration that chronically stimulates the DA receptor in the postganglionic neuron. Furthermore, cocaine binds to the 5-HT transporter and inhibits 5-HT reuptake [5]. The “desired effects” of cocaine are mainly related to the increased extracellular dopamine concentration in the synaptic cleft, and include euphoria, psychic energy, enhanced sexual excitement, and self-confidence [6,7,8].

Cocaine administration leads to a multiorgan toxicity due to its redistribution through the bloodstream [8]. The literature is full of studies on cocaine cardiotoxicity [9,10,11,12], but consequences involving the brain, lungs, kidneys, the gastrointestinal tract, musculature, and sexual organs are also well documented [13,14,15,16,17,18]. Peripherally, the increased concentration of catecholamines at the synapse leads to α- and β-adrenergic stimulation. In the heart tissue, in addition to its potent sympathomimetic effect, cocaine exerts an anesthetic effect by blocking sodium channels. This leads to an alteration of cellular calcium homeostasis, and an elevated myocardial oxygen demand, paralleled by a concomitant decrease in oxygen supply, causing an impaired cardiac functionality [11,19]. Despite the knowledge of the triggering components, the mechanisms of cocaine-induced cardiotoxicity and the subcellular targets are still under investigation [17]. 

Forensic medicine has always shown particular attention to the morpho-pathological aspects of cocaine-related deaths. Since the early 2000s, the molecular pathophysiological mechanisms of cocaine have captured interest as alternative lethal aspects in addition to acute intoxication [7,20,21]. In this context, a large body of evidence highlights the role of oxidative stress in the resulting multi-toxic effects of cocaine, with great deal on mitochondrial impairment [22,23,24,25,26,27,28,29,30,31,32]. 

The mitochondrion is considered a central organelle in the pathophysiology of cells and tissues. It is a double membrane organelle with its own DNA, which is not only the power house that produces adenosine triphosphate (ATP) for the cell, but also a central coordinator in cell processes such as energy sensing, reactive oxygen species (ROS) and reactive nitrogen species (RNS) equilibrium, and apoptosis contributing to cell stress regulation [33]. Mitochondrial dynamics are quite articulated; mitochondria can undergo fission, fusion, and mitophagy, controlling their number and activity, but also participating in the regulation of cell death [33].

The present paper aims to review the literature on mitochondrial injury and oxidative stress underlying cocaine cardiotoxicity. The review initially provides a summary of the functional and morphological aspects crucial for the medical and forensic aspects, to then connect to molecular evidence that highlights the central role of mitochondria in cocaine cardiotoxicity. Considering the paucity of data on cocaine-induced mitochondrial damage in humans, forensic material represents a precious source to study the molecular aspects of cocaine cardiotoxicity. This will contribute to the determination of predisposing factors and to finding strategies to reduce cocaine’s impact on mitochondrial integrity.

## 2. Results

### 2.1. Literature Search Results

Overall, 106 articles were found, and 54 duplicate records were removed. Afterwards, the remaining 52 articles were subjected to the screening phase during which 9 articles were excluded by title review and 30 by abstract review. A total of 13 were assessed for eligibility and, after full-text review, only 5 articles met the inclusion criteria. In addition, 17 articles were found after searching among those reported in bibliographies and 9 articles were selected. Finally, a total of 14 articles were included in the study. The complete flow chart is shown in Figure 1.

The literature review has allowed for us to identify the presence of 14 articles that specifically explored the cellular mechanisms of mitochondria-mediated cocaine cardiotoxicity. Articles included in the study are listed in Table 1.

### 2.2. Characteristics of the Included Articles

The preliminary analysis of the articles revealed heterogeneity in the experimental models employed and the parameters analyzed. In four articles, the authors included an in vivo assessment of cocaine abuse in an animal model [38,39,41,43]. The animals were tested either after the infusion of cocaine [38] or after the combined infusion of cocaine and a specific treatment [39,41,43], and the parameters indicative of cardiac dysfunction were explored. Eight articles included ex vivo examination of the heart from animal models after the administration of cocaine [38,39,40,41,42,43,46,47], and six articles provided an in vitro study on cell cultures [34,35,36,37,38,45]. Only one study investigated human heart tissue though paraffin-embedded specimens collected from cocaine-related deaths [44]. The research papers analyzed a variety of parameters, spanning from physiological to molecular markers, which correlate with cardiac dysfunction and are associated with mitochondrial damage and oxidative stress. Figure 2 summarizes the characterized mitochondrial targets, the resulting effects on mitochondrial functions, and the altered cardiac structure and function.

## 3. Discussion

### 3.1. Morpho-Functional Parameters to Assess Cocaine Cardiotoxicity

Cocaine cardiotoxicity can be measured in different ways, spanning from the survey of functional parameters of the organ to the assessment of its morphological characteristics. 

One of the functional parameters to assess cocaine-induced cardiac dysfunction consists of the evaluation of left ventricular (LV) relaxation. All the in vivo studies reported that the administration of an adequate dose of cocaine led to cardiac impairment. According to Lattanzio et al. [38], the effects of the exposure to a single dose of cocaine caused a modest reduction in LV cardiac performance in animal models, which included an impaired contraction, alterations in relaxation rate and duration, and a reduced heart rate. The molecular mechanisms underlying cocaine-mediated LV modifications were investigated using cardiac cell lines, suggesting that the pathological mechanisms included an altered depolarization of mitochondrial membrane and an increase in oxidative stress and calcium concentration in mitochondria and cytoplasm [38]. The effects of continuous cocaine administration have been investigated as well. In a seven days administration protocol, Vergeade et al. [41] reported the onset of cardiac dysfunction with an increase in LV relaxation, LV end-diastolic pressure-volume relation, and LV end-diastolic pressure. 

Cardiac damage can be routinely investigated by histological examinations. Accordingly, five studies evaluated the presence of changes in cardiomyocytes morphology of both tissue origin and culture [34,36,37,42,46,47]. In the analysis performed seven days after the administration of cocaine via the rat tail vein, Wen et al. [46] found no signs of hypertrophy, nor the presence of inflammatory infiltrates or fibrosis deposition in the myocardium, suggesting the absence of macroscopical alterations after a week administration of the drug. Intriguingly, another study performed by the same group, revealed a striking difference when the treatment period was extended from 7 to 14 days [47]. In this condition, the histopathological examination of the myocardium revealed the presence of contraction band necrosis together with eosinophil infiltrations, focal myocytolysis, swelling of cardiomyocytes nuclei, and abundant perivascular collagen deposits [47]. Liou et al. [42], who performed a three-month cocaine treatment on rats, reported a consistent abnormal myocardial architecture and revealed an enhanced cardiac apoptotic activity, as demonstrated by the increased number of TUNEL-positive cardiac cells. As a confirmation of the direct damage of cocaine to cardiomyocytes, Xiao et al. described features ascribable to apoptosis [36]. 

Interestingly, cocaine cardiotoxicity has been correlated with the impairment of other markers such as Adiponectin, Annexin V, lactate dehydrogenase (LDH), and Connexin-43 (Cx43) [48,49,50,51]. Adiponectin, Annexin V, and LDH are considered as tools to measure the presence of general toxicity that leads to apoptosis, necrosis, and cell death, while Cx43 is a less investigated marker of cardiotoxicity [48,49,50]. Cx43 belongs to a family of widely expressed trans membrane glycoproteins, which in the cardiomyocytes is the most important protein constituting the gap junctions at intercalated discs level [51]. Under normal conditions, Cx43 is in a phosphorylated state, but in pathological situations Cx43 undergoes a dephosphorylation, resulting in an altered intercellular communication and enhancing myocardial apoptosis [51]. Wen et al. [47] found a significant increase in dephosphorylated Cx43 at Serine 368 at the intercalated discs in the myocardial tissue of a cocaine abuse rat model, compared to control. Moreover, Cx43 has been suggested to be essential for the regulation of cardiomyocytes homeostasis, oxygen consumption, the regulation of potassium fluxes, and oxidative stress response, therefore concurring to cardio-protection [51,52]. 

### 3.2. Loss of Mitochondrial Homeostasis in Cocaine Cardiotoxicity 

As previously anticipated, the mitochondrion is considered a central organelle in cell and tissue pathophysiology and, to date, it is considered a sensor of cellular stress [33,53,54,55]. The experimental contributions reviewed here report different mitochondrial damages that concur to cocaine cardiotoxicity, which will be discussed below.

The primary function of mitochondria is to produce ATP [33], which is a key factor in a high energy demand organ, such as the heart is. Therefore, the maintenance of ATP production is an extensively investigated aspect in the included articles (see Table 1). The effects of cocaine on cardiac mitochondrial energetic function could be mediated by a direct alteration in respiratory chain apparatus and its indirect damage via oxidative stress [32]. In this context, Vergeade et al. [39] observed an impaired ATP production and oxygen consumption through complex I and III, which are also major productors of ROS. In a later study, Vergeade et al. [41] confirmed the major role of complex III in the alteration in oxidative state induced by cocaine by testing amytal (complex I inhibitor) and antimycin A (complex III inhibitor). On the contrary, Wen et al. [47] found an elevation in the generation of ROS due to an increase in complex I functionality. These discrepancies can be attributable to the different experimental designs, or to a complex cellular condition. In addition to the correct functionality of the respiratory complexes, the mitochondrial membrane potential (ΔΨ) plays a fundamental role in the activity of the respiratory chain and in cell survival. Yuan et al. [34] suggested a dose- and time-dependent relation between cocaine exposure and mitochondrial ΔΨ dissipation. 

Moreover, signs of calcium overload such as an altered shape of mitochondria and the presence of regions of higher density, which are signs of calcium overload, were observed after cocaine exposure for 14 days in an ex vivo experiment [34]. Lattanzio et al. [38] used confocal microscopy with fluorescent calcium indicators fluo-3 (cytoplasmatic sites) and rhod-2 (mitochondrial sites) to estimate changes in calcium levels in LV cardiomyocytes with increasing concentrations of cocaine, finding a dose-dependent increase in both signals. The consequences of the impairment of calcium homeostasis include changes in ΔΨ, thus leading to ATP depletion, an excessive production of ROS, and apoptosis [56], which also characterizes other morbid states such as ischemia and infarction [54].

Several works have described a cocaine-dependent role in oxidative stress [32,57,58,59,60,61,62,63]. Interestingly, treatment with allopurinol—a xanthine oxidase inhibitor—and MitoQ—a mitochondrial-specific antioxidant—seven days after cocaine consumption in an animal model prevented mitochondrial alterations, thus supporting the role of oxidative stress in cocaine-induced cardiotoxicity [39,41]. Fettiplace et al. [43] demonstrated that cocaine potently inhibited mitochondrial carnitine exchange and suggested the carnitine-acylcarnitine translocase (CACT) inhibition as a mechanism of toxicity. It is worth noting that CACT allows for mitochondrial respiration by catalyzing the transport of short, medium, and long carbon chain acyl-carnitines through the inner membrane of mitochondria, by exchanging free carnitine [64]. It was suggested that the exposition to carnitine could bypass the inhibition of pyruvate dehydrogenase caused by cocaine, thus sustaining the production of nicotinamide adenine dinucleotide (NADH) through acetyl-CoA supply [43]. Moreover, Fettiplace et al. [43] showed that a pre-treatment with intravenous lipid emulsion was able to delay cardiotoxicity, defined as systolic pressure below 40 mmHg (cut-off considered as at risk of ischemic sequelae) and the development of asystole. Therefore, lipid emulsion seems to improve mitochondrial lipid metabolism counteracting cocaine-induced cardiotoxicity [43]. Additionally, Lattanzio et al. [38] found a significant mitochondrial membrane depolarization of LV cardiomyocytes treated with 10^−2^–10^−4^ M concentrations of cocaine, which was attenuated by pretreatment with N-acetylcysteine (NAC), a free radical scavenger. Lastly, RNS can also contribute to cocaine-dependent cell-stress, as demonstrated by Turillazzi et al. [44]. This study was performed on human cardiac specimens from cocaine overdose victims. It revealed an increase in the expression of nicotinamide adenine dinucleotide phosphate oxidase 2 (NOX2), inducible nitric oxide synthase (i-NOS), and nitrotyrosine compared to the control group.

In this context, it is also worth considering glutathione metabolism, which is the most important physiological scavenger of ROS and RNS. In particular, the ratio between reduced glutathione (GSH) and oxidized glutathione (GSSG) may be used as a marker of oxidative stress [65]. In particular, GSH is a cytosolic redox buffer essential for cells’ protection against redox injury, which works in synergy with ascorbic acid (AA), both at the level of the cytosol and mitochondria [66]. Actually, the imbalance in the levels of GSH/GSSG ratio and AA makes the cell susceptible to free radical stress [65]. In their in vitro study, Martins et al. [45] found a mild depletion in cytosolic GSH and an increase in GSSG, especially at the highest concentration of cocaine. Comparable results were observed in animal models. In particular, Sinha-Hikim et al. [40] described a decrease in GSH/GSSG ratio and significant high levels of 4-hydroxy-trans-2-nonenal (4-HNE), which is a marker of an increased lipid peroxidation [67]. Interestingly, the authors showed that continuous treatment with minocycline prevented oxidative stress in the fetal heart [40], probably through the inhibition of the mitochondria-dependent death pathway [68,69]. Moreover, the increase in 4-HNE concentrations was also reported by Wen et al. [47], as well as higher levels of 8-hydroxy-2′-deoxyguansine (8-OHdG), a marker of effects of oxidative stress on DNA [70]. In addition, Turillazzi et al. [44] showed that the GSH/GSSG ratio and AA levels were decreased, and the levels of malondialdehyde (MDA), another indicator of lipid peroxidation [67], were significantly higher in individuals who died for cocaine overdose. Furthermore, the authors reported similar results to Wen et al. in terms of 8-OHdG presence, which was found to be selectively expressed in cardiomyocytes’ nuclei [44,47]. 

Based on the literature data, calcium overload leads to the mitochondrial permeability transition pore (mPTP) opening that, as reported by Martins et al. [45], contributes to the impairment of mitochondrial membrane permeability after 24 h exposure to cocaine and the release of pro-apoptotic factors. When mitochondria are not able to buffer stress insults, they contribute to apoptosis, leading to cell death. Apoptosis is regulated via an extrinsic pathway, initiated by extracellular stimuli, or via an intrinsic pathway, induced by intracellular events [71,72]. Although with different modalities and molecular actors, both pathways can involve the activation of caspases and the participation of mitochondria. 

Li et al. [37] showed that, in vitro, cocaine induces a time-dependent cytochrome c release, which precedes cocaine-induced cell apoptosis. Moreover, Xiao et al. [36], working with cell cultures, found a threefold increase in cytosolic cytochrome c, and a mitochondrial decrease of about 60% due to cocaine exposure. In addition, the authors reported a dose-dependent effect of cyclosporin A, an inhibitor of apoptosis, finding a decrease in the translocation of cytochrome c in the cytosolic space [36]. Similar results were obtained in animal models exposed to cocaine followed by continuous treatment with minocycline, another inhibitor of the cytosolic translocation of cytochrome c and the mitochondria-dependent pathway of death [40]. Furthermore, Liou et al. [42] found a significant increase in cytosolic cytochrome c in a model of chronic abuse. Therefore, this data confirms the central role of cocaine in mitochondria-regulated cell death.

However, the complexity of the apoptotic pathway [71], together with the broad spectrum of cocaine effects on tissues and cells, leads us to hypothesize that its toxicity is not limited to the intrinsic apoptotic pathway, but probably extends to the alteration in different molecular mechanisms. Therefore, in order to understand the pro-apoptotic mechanisms induced by cocaine, several works have exploited both in vitro and in vivo models to investigate the level of activation of caspase 8 (Fas-dependent—extrinsic apoptotic pathway), caspase 9 (mitochondria-dependent—intrinsic apoptotic pathway) and caspase 3 (common apoptotic pathway), BH3 interacting domain death agonist (Bid), B-cell lymphoma-2 (Bcl-2), and second mitochondria-derived activator of caspase or direct inhibitor of apoptosis binding protein with low isoelectric point (Smac/DIABLO). 

An in vitro study on cell lines showed an increase in the activity of caspase 9, 8, and 3 following cocaine exposure [45]. While experiments on fetal rat myocardial primary cells culture showed an increase in the activity of caspase 9 and 3, no significant changes in caspase 8 activity were observed [37]. Accordingly, Sinha-Hikim et al. [40] showed that in utero cocaine administration induced caspase 9 and 3 upregulation, which was attenuated by the concomitant administration of minocycline. Due to the contrasting results, the contribution of extracellular stimuli to apoptosis has been further investigated. Specifically, tumor necrosis factor-α (TNFα), Fas, Fas ligand (FasL), and Fas-associated death domain (FADD) represent the upstream components of the cardiac extrinsic pathways of apoptosis. TNFα and FasL may both initiate the apoptotic pathway. While the interaction of FasL with the Fas-receptor leads to the direct death-inducing signal complex with FADD, the TNFα pathway requires the formation of an oligomer with its receptor and an adapter protein for the recruitment of FADD [72]. These processes will determine the activation of caspase 8 [71]. Both cell cultures and LV cells from rabbits displayed an increase in the expression of genes of the extrinsic apoptotic pathway such as FasL, Fas, and TNFα [38]. Similarly, a study on the hearts of rats previously treated with cocaine found increased levels not only of TNFα, FasL, and Fas, but also FADD [42]. Moreover, analyses on human cardiac tissue of cocaine-related overdoses found a significant increase in TNFα immunoexpression, compared to the control group [44].

Evidence suggests that cocaine-induced cardiotoxicity includes the alteration of further molecular mechanisms involved in the regulation of apoptosis. In their studies with animal models, Wen et al. and Liou et al. [42,47] found a significant increase in pro-apoptotic proteins such as Bcl-2-associated X protein (Bax) and Bcl-2 homologous antagonist/killer (Bak) in the cocaine group, thus inducing changes in mitochondrial membrane permeability and the release of pro-apoptotic proteins [73]. Moreover, Liou et al. [42] reported lower levels of Bcl-xL, an anti-apoptotic protein, in the presence of cocaine. Although Li et al. [37] found a reduction in the levels of Bcl-2, another anti-apoptotic protein, no difference in Bax concentrations was observed after cocaine exposure. Interestingly, Liou et al. [42] also reported a significant increase in the truncated Bid (t-Bid)/Bid ratio. Bid is an intracellular mediator that connects the Fas-dependent apoptosis pathway to the mitochondria-dependent apoptosis pathway [73]. Specifically, Bid is cleaved by caspase 8 into t-Bid, which oligomerizes with Bak causing its activation and the release of cytochrome. In addition, Turillazzi et al. [44] suggested a cocaine-dependent increase in Smac/DIABLO, which has similar properties as cytochrome c once it is released from the mitochondria [74]. 

Not least, according to the observation that mitochondria biogenesis contributes to apoptosis suppression [75], Wen and collaborators reported an alteration in mitochondrial dynamics markers, namely, a short-term cocaine administration induced a reduction in mitochondrial fission 1 protein (Fis1), Mitofusin 2 (Mfn2), and Optic atrophy type 1 (Opa1) [46,47].

To resume, cocaine-induced apoptosis can be mediated by both the extrinsic and intrinsic pathway, differently involving the mitochondria, and providing to these organelles a central role in cocaine cell toxicity. The reported studies present heterogeneity in terms of experimental conditions and models, and consequent responses. Therefore, cocaine cardiotoxicity probably affects cardiomyocyte function by activating multiple pathways that entail mitochondria-dependent cell death [76].

## 4. Materials and Methods

### 4.1. Search Strategy

A comprehensive literature search about cocaine-induced mitochondrial damage-related cardiotoxicity was made on Pubmed up to 1 May 2023, following the Preferred Reporting Items for Systemic Reviews and Meta-Analysis (PRISMA) statement’s criteria in accordance with PRISMA 2020 guidelines. The authors searched the available English-languages articles; no time limits for publication were set. In the identification phase, a combination of the terms “mitochondrial damage” or “mitochondrial injury” and “cocaine” and/or “cardiac” and/or “abuse” were searched in every field. 

### 4.2. Inclusion Criteria

To achieve the purpose of the review, the authors included published research papers in English that focused on: (i) effects of cocaine on cardiac mitochondria; (ii) molecular pathways of cocaine cardiotoxicity; and (iii) possible treatment of cocaine-induced oxidative stress.

### 4.3. Exclusion Criteria 

The authors did not include review articles, pre-proof articles, and articles that were not directly related to cocaine-induced mitochondrial damage-related cardiotoxicity. Published papers that did not considered cocaine effects, but other substances, were excluded. Articles that take into account cocaine-induced mitochondrial damage-related nephrotoxicity, hepatotoxicity, and neurotoxicity were not included.

## 5. Conclusions

This manuscript reports the broad-spectrum effects of cocaine on cardiac mitochondria. The above plethora of impaired mitochondrial functions in cardiomyocytes provides this organelle with a central role in the development of cocaine-induced cardiotoxicity. This systematic review highlights a knowledge gap in this research field due to the limited number of research publications spread over a wide time window. Despite the important role of mitochondria, the molecular pathways responsible for their dysfunction are still poorly investigated. Therefore, further studies are needed to improve our understanding of the complex processes involved, which will also be crucial to identifying potential therapeutic targets to mitigate or reverse cocaine toxicity. Moreover, considering that experimental models do not fully reflect cellular pathways in the human body, forensic material represents a valuable source to study the molecular aspects of cocaine cardiotoxicity, ideally contributing to the determination of predisposition factors and finding strategies to reduce the impact of cocaine on the heart.

## Figures and Tables

**Figure 1 ijms-24-14517-f001:**
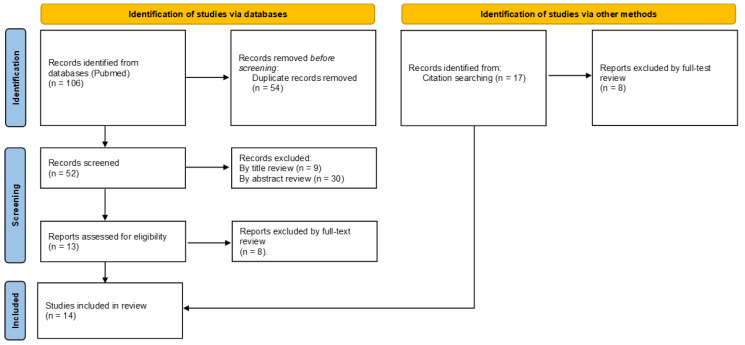
Flow diagram showing a summary of the search strategy used regarding cocaine-induced mitochondrial damage-related cardiotoxicity in accordance with PRISMA 2020 guidelines. In the identification phase a total of 106 articles were identified. Removal of duplicates and exclusion criteria allowed for the final inclusion of 5 articles. Further, 9 articles were added as identified by citations.

**Figure 2 ijms-24-14517-f002:**
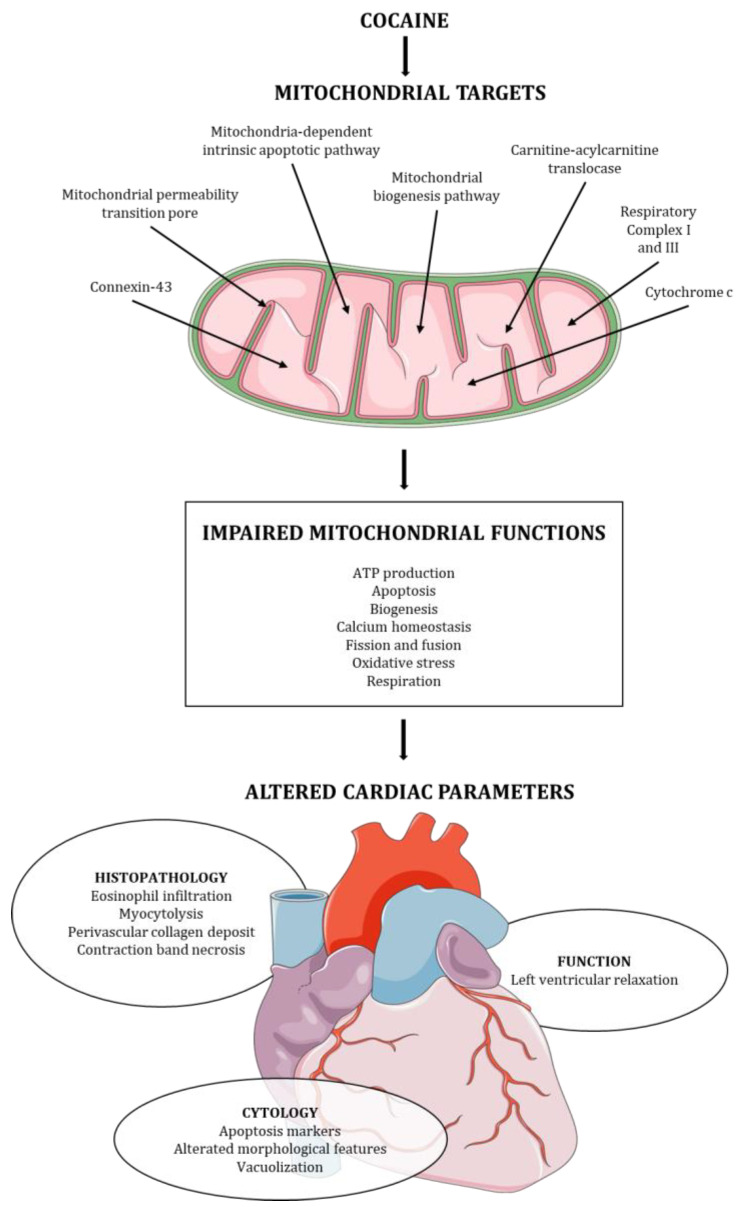
Cocaine mitochondrial targets, mitochondrial functions affected, and cardiac structure and function alteration induced by cocaine.

**Table 1 ijms-24-14517-t001:** List of the selected articles reporting experimental conditions and affected mitochondrial functions observed.

Articles	Experimental Model	Cocaine Treatment	Main Findings	Affected Cellular Functions
Yuan et al., 1996 [34]	Primary rat myocardial cells	3 to 48 h, 10^−5^ to 10^−3^ M	ΔΨ dissipation Complex I inhibition Altered NADH-linked pathway	Respiration
		Repeated exposure: 24, 48, 72 h, 10^−6^ to 10^−3^ M		
Yuan et al., 2000 [35]	Primary neonatal rat myocardial cells	6, 12, 24 h, 10^−5^ to 10^−3^ M	Complex I inhibition Altered NADH-linked pathway	Respiration
Xiao et al., 2000 [36]	Primary fetal rat myocardial cells	24 h, 100 µM	Cytochrome c release Caspase 3 and 9 activation	Apoptosis
Li et al., 2005 [37]	Primary fetal rat myocardial cells	3 to 48 h, 100 µM	p38α-MAPK activation Caspase 3 and 9 activation	Apoptosis
Lattanzio et al., 2005 [38]	New Zealand White female rabbits, in vivo and ex vivo hearts	Single dose, intravenous injection, 2 mg/kg	Increased free radicals Altered intracellular calcium Apoptotic genes activation	Oxidative stress Apoptosis
	H9c2 rat cardiac cells	2 to 5 min, 10^−5^ to 10^−2^ M		
Vergeade et al., 2010 [39]	Wistar male rats, in vivo and ex vivo hearts	2 to 7 days, intraperitoneal injection, 2 × 7.5 mg/kg/day	Increased mitochondrial ROS Complex I and III alteration	Oxidative stress Respiration
Sinha-Hikim et al., 2011 [40]	Ex vivo newborn hearts from Sprague Dawley pregnant rats	E15 to E21 intraperitoneal injection to pregnant rats, 2 × 15 mg/kg/day	p38α-MAPK activation JNK activation Altered GSH/GSSG ratio Cytochrome c release Caspase 3 and 9 activation	Oxidative stress Apoptosis
Vergeade et al., 2012 [41]	Wistar male rats, in vivo and ex vivo hearts	7 days, intraperitoneal injection, 2 × 7.5 mg/kg/day	Xanthine oxidase-dependent mitochondrial ROS production	Oxidative stress Respiration
Liou et al., 2014 [42]	Wistar male rats, ex vivo hearts	3 months, subcutaneous injection, 10 mg/kg/day	Altered Bax/Bcl2 ratio Cytochrome c release	Apoptosis
Fettiplace et al., 2015 [43]	Sprague Dawley rats, in vivo and ex vivo hearts	Continuous infusion, 10 mg/kg/min	Altered CACT-dependent respiration	Respiration
Turillazzi et al., 2017 [44]	Heart specimens from human cocaine-related overdoses	Cocaine overdoses	Increased i-NOS Altered GSH/GSSG ratio Increased SMAC-Diablo pathway	Oxidative stress Apoptosis
Martins et al., 2018 [45]	H9c2 rat cardiac cells	24 h, 104 µM to 6.5 mM	ΔΨ alteration Increased free radicals Altered GSH/GSSG ratio Alteration of Caspase 8, 9 and 3	Oxidative stress Apoptosis Respiration
Wen et al., 2021 [46]	Sprague Dawley male rats, ex vivo hearts	7 days, intravenous injections, 5 mg/kg/day–20 mg/kg/day	Alteration of Fis1, Mfn2, Opa1 and PPARα Complex I inhibition UPRmt activation	Mitochondrial dynamics Respiration
Wen et al., 2022 [47]	Sprague Dawley male rat, ex vivo hearts	14 days, intravenous injection, 20 mg/kg/day	Activation of Akt and Bax/Bcl2 Complex I alteration Altered Caspase 3 Increase in 4-HNE and 8-OHdG conjugates	Oxidative stress Apoptosis

## Data Availability

Not applicable.

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
