# Peer review of "Subcellular Effectors of Cocaine Cardiotoxicity: All Roads Lead to Mitochondria—A Systematic Review of the Literature"

_ijms, 2023, doi:10.3390/ijms241914517_

Round 1
Reviewer 1 Report
The review by Peruch et al. is a well-written review summarizing the correlation between cocaine cardiotoxicity and mitochondrial damage. The authors presented articles that studied the cellular mechanisms of mitochondria-mediated cocaine cardiotoxicity. The studies include impaired energy production, membrane potential imbalance, increased oxidative stress and apoptosis. This review is discussing an interesting topic.
Comments
1-Line 103 has a comment that needs to be removed.
2-Line 196 has a comment that needs to be removed and the missing reference should be added.
3-Line 290 the reference 40 is not the correct reference and should be replaced by reference 43.
Author Response
We would like to express our appreciation for the positive feedback regarding our manuscript.
- Line 103 has a comment that needs to be removed.
We thank the Reviewer for the comment, the comment has been removed and the citations included accordingly (line 138).
- Line 196 has a comment that needs to be removed and the missing reference should be added.
As suggested by the Reviewer, the comment has been removed and the missing reference added (line 233).
- Line 290 the reference 40 is not the correct reference and should be replaced by reference 43.
We thank the Reviewer for the comment, the reference has been replaced (line 326).
Reviewer 2 Report
The article "Cocaine Cardiotoxicity: All Roads Lead to Mitochondria. Review of the Literature" by Michela Peruch et al. is an interesting, important and relevant study of the mechanisms by which cocaine cardiotoxicity affects mitochondrial functioning. The authors have tried to link important issues of cocaine administration with elements of forensic medicine and biology, which, in principle, does not correspond to the journal's topic and the concept of the investigated subject.
A close examination of the text reveals several important points of material presentation that significantly compromise the value of the presented work.
First of all, the title should be changed, as the clarification of the research topic, namely related to the literature review, and it will be done in the abstract.
The abstract itself is written chaotically and does not represent the novelty of the carried out research, does not contain information about the used sources of literature analysis and their scope. Such presentation of material is unacceptable for this type of articles.
The authors have no graphical material in their work, which represents the description of the processes under study, and the presence of only one table is not enough for this type of work.
The author should include the literature search strategy for review, which would help to present the work better.
There are great remarks concerning the table, which is made very superficially and in no way represents the mechanisms and effects of cocaine intoxication. This refers primarily to the structure of the table and the details of the effects, which do not represent the molecular mechanisms of the effects. For the journal level regarding molecular processes the descriptions given by the authors as "oxidative stress", "Apoptotic pathway" and "ATP production" are insufficient and do not reflect the molecular mechanisms of the realized processes.
The article is written with spelling errors and stylistically unclear approaches, which indicates that the work was not carefully prepared for submission. For example, "Calcium omeostasis" in the table, and in line 103-104 "In four articles (which? Cite them), the authors included in vivo assessment of cocaine abuse in an animal model", etc. The article is not carefully prepared.
Perhaps, for this type of work related to literature review and not of experimental nature, as the authors declare, the division of the work into Results and Discussion sections is unreasonable. The titles of the subsections are too long.
For innovative type of work, to which this paper can hardly be referred, it is necessary to use literature mainly from the last 5-10 years, but the authors do not have this. This also reduces the value of the research value of the paper.
The work requires complete revision and in this form cannot be considered for publication.
Author Response
First of all, we would like to thank the Reviewer for the comments regarding our manuscript and all the valuable suggestions.
- The article "Cocaine Cardiotoxicity: All Roads Lead to Mitochondria. Review of the Literature" by Michela Peruch et al. is an interesting, important and relevant study of the mechanisms by which cocaine cardiotoxicity affects mitochondrial functioning. The authors have tried to link important issues of cocaine administration with elements of forensic medicine and biology, which, in principle, does not correspond to the journal's topic and the concept of the investigated subject.
The current manuscript has been submitted to the special issue “Mitochondria-Mediated Oxidative Stress in Diseases: Cell Death and Treatment” after invitation of Prof. Dr. Renata A. Zvyagilskaya.
Actually, the authors aimed at reviewing the literature on the molecular pathways underlying mitochondrial cocaine toxicity. In this systematic review, the need to undertake targeted studies on human tissue samples has emerged. Therefore, in a future perspective, the deep knowledge of cocaine cardiotoxicity could be of crucial help to find strategies for novel treatments.
To make this concept clear we modifies lines 357-363.
- First of all, the title should be changed, as the clarification of the research topic, namely related to the literature review, and it will be done in the abstract.
With this title the authors want to highlight the central role of mitochondria. We believe that, together with the changes made to the text thanks to the Reviewer’s suggestions, the title summarizes the message of the current review.
- The abstract itself is written chaotically and does not represent the novelty of the carried out research, does not contain information about the used sources of literature analysis and their scope.
We appreciate the suggestion of the reviewer and we extensively changed the abstract to solve the highlighted criticisms. Lines 15-30.
- The authors have no graphical material in their work, which represents the description of the processes under study, and the presence of only one table is not enough for this type of work.
Following the Reviewer’s constructive comment, we added the figure 1 to describe the workflow of the study. Moreover, we included an additional figure (figure 2) to better clarify the mitochondrial molecular mechanisms involved in cocaine cardiotoxicity.
- The author should include the literature search strategy for review, which would help to present the work better.
The authors thank the Reviewer for rising this point and we apologize for the inaccuracy. However, the manuscript was formatted not including this information after the correction of the Assistant Editor, who suggested to “…remove the method section”.
The search strategy, inclusion and exclusion criteria have now been included in the “Materials and Methods” paragraph. Lines 91-110.
- There are great remarks concerning the table, which is made very superficially and in no way represents the mechanisms and effects of cocaine intoxication. This refers primarily to the structure of the table and the details of the effects, which do not represent the molecular mechanisms of the effects. For the journal level regarding molecular processes the descriptions given by the authors as "oxidative stress", "Apoptotic pathway" and "ATP production" are insufficient and do not reflect the molecbular mechanisms of the realized processes.
We thank the Reviewer for the comment. With Table 1, we aimed to reflect the heterogeneity of methods in the included research publications. Therefore, we believe that the Figure 2 could summarize the molecular mechanism of cocaine on mitochondria. As we pointed in the conclusion (lines 358-359), we observed a lack in this research field. Actually, most of the papers report mainly alteration to mitochondria functions and only a few delve into molecular mechanism.
- The article is written with spelling errors and stylistically unclear approaches, which indicates that the work was not carefully prepared for submission. For example, "Calcium omeostasis" in the table, and in line 103-104 "In four articles (which? Cite them), the authors included in vivo assessment of cocaine abuse in an animal model", etc.
We thank the Reviewer for the comment. The text has been edited and the citations included.
- Perhaps, for this type of work related to literature review and not of experimental nature, as the authors declare, the division of the work into Results and Discussion sections is unreasonable. The titles of the subsections are too long.
As answered in point 5, the authors introduced a section for the search strategy and results. Moreover, discussion subheadings have been rephrased at lines 153 and 203.
- For innovative type of work, to which this paper can hardly be referred, it is necessary to use literature mainly from the last 5-10 years, but the authors do not have this. This also reduces the value of the research value of the paper.
The authors understand the concerns of the Reviewer. Actually, we are aware of the scarcity of data presented in literature and, indeed, we would like to underline that observations reported are rather dated. Given the profound public health implications of cocaine cardiotoxicity, we would like to emphasize that more research needs to be done. Lines 361-367.
Round 2
Reviewer 2 Report
Dear Editor,
The paper "Cocaine Cardiotoxicity: All Roads Lead to Mitochondria. Review of the Literature" by Michela Peruch et al. submitted for review is an interesting study of pathophysiological mechanisms of cardiotoxicity and other tissues induced by cocaine derivatives.
The authors made some minor changes in the text according to the reviewers' comments.
However, the authors did not make the main corrections of their text, which make it impossible to accept this work in this form for publication.
The high level of requirements for publications in this journal requires demonstration of molecular mechanisms of cocaine effects realization for clinical research practice, which is not shown by the authors (see summarizing Table 1), so this approach of the authors significantly reduces the value of the presented work.
First of all, this refers to the summarizing table, which does not include an additional column elucidating the molecular mechanisms of cocaine effects. The table contains the terms "Oxidative stress", "ATP production", "Apoptotic pathway", etc., which are very generalized, relate to many pathological mechanisms, and, accordingly, do not elucidate the specific mechanisms of cocaine action. Importantly, this is unacceptable for review articles in a journal of this level. It reflects too superficially the problem under study or represents the authors' incompetence in the subject.
It should be noted that clarification of the molecular mechanisms of cocaine action involves receptor control, and this issue was completely missed by the authors. This only confirms the non-careful manner in which the study was approached.
The caption to Figure 1 does not provide information regarding the subject of the literature search and does not clearly present the search strategy.
The title of the paper is unspecific and very superficial.
The paper requires careful revision and more involvement of the authors in the study of the molecular mechanisms of cocaine effects, which are presented primitively and very superficially, which does not meet the high criteria of the journal.
Author Response
Dear Reviewer, thank you for your suggestions.
The paper "Cocaine Cardiotoxicity: All Roads Lead to Mitochondria. Review of the Literature" by Michela Peruch et al. submitted for review is an interesting study of pathophysiological mechanisms of cardiotoxicity and other tissues induced by cocaine derivatives. The authors made some minor changes in the text according to the reviewers' comments. However, the authors did not make the main corrections of their text, which make it impossible to accept this work in this form for publication. The high level of requirements for publications in this journal requires demonstration of molecular mechanisms of cocaine effects realization for clinical research practice, which is not shown by the authors (see summarizing Table 1), so this approach of the authors significantly reduces the value of the presented work. First of all, this refers to the summarizing table, which does not include an additional column elucidating the molecular mechanisms of cocaine effects. The table contains the terms "Oxidative stress", "ATP production", "Apoptotic pathway", etc., which are very generalized, relate to many pathological mechanisms, and, accordingly, do not elucidate the specific mechanisms of cocaine action. Importantly, this is unacceptable for review articles in a journal of this level. It reflects too superficially the problem under study or represents the authors' incompetence in the subject.
Table 1 has been modified to meet the Reviewer comments.
The Authors would like to highlight that in some cases (see for example Yuan et al, 1996), the studies reported still quite general mechanisms (dissipation of mitochondrial membrane potential, complex I inhibition, et cetera), rather than detailed molecular pathways. This is not dependent on superficiality, but relays on the evidence that at the beginning of the 2000s, the role of mitochondria, and the related pathways, in cocaine cardiotoxicity, was still quite unexplored. And, only recently, more in-depth information is issuing from a quite restricted number of research articles (see Table 1). Authors attempted to elaborate a concise and explicative table. In parallel, the Reader will find all the relative details in the main manuscript. It should be noted that clarification of the molecular mechanisms of cocaine action involves receptor control, and this issue was completely missed by the authors. This only confirms the non-careful manner in which the study was approached.
Authors are aware that cocaine cardio-toxicity is mainly ascribable to an augmented adrenergic activity that increases the myocardial contractility and conduction, together with a local anesthetic effect that is caused by inhibition of the transient inward flux of sodium across the cell membrane during depolarization. However, recent literature agrees on the evidence that cardiac effects of cocaine are multifaceted. Altogether, the mechanisms of cocaine-induced cardiotoxicity remain largely unclear, and the subcellular targets of cocaine have not been elucidated in myocardial cells. Although receptors control is the triggering point, other factors such as tissue inflammation, oxidative stress, intracellular calcium homeostasis, and in general cell stress, participate to cocaine cardiotoxicity. Without taking dignity away from the receptor control concept, this review wants to delve into one of the most important cellular organelles, the mitochondrion, which is considered a cellular stress sensor. This has been motivated by the evidence that cocaine cardiotoxicity entails damages to the myocardial cells correlated to mitochondrial functioning. This concept has been better developed in the Introduction (see lines 56-83).
The caption to Figure 1 does not provide information regarding the subject of the literature search and does not clearly present the search strategy.
Figure 1 caption has been modified according to the Reviewer’s comments.
The title of the paper is unspecific and very superficial.
According to the answers to the previous points, and to meet the Reviewer comments the title has been changed in the following:
“Subcellular effectors of cocaine cardiotoxicity: all roads lead to mitochondria. A systematic review of the literature.”

Round 3
Reviewer 2 Report
Dear Editor,
The authors have made all necessary corrections to the text of the article. The article may be accepted for publication in this form.